# TIGHTER SPARSE APPROXIMATION BOUNDS FOR RELU NEURAL NETWORKS

## ABSTRACT

A well-known line of work (Barron, 1993; Breiman, 1993; Klusowski & Barron, 2018) provides bounds on the width $n$ of a ReLU two-layer neural network needed to approximate a function $f$ over the ball $\mathcal{B}_R(\mathbb{R}^d)$ up to error $\epsilon$, when the *Fourier* based quantity $C_f = \int_{\mathbb{R}^d} \|\xi\|^2 |\hat{f}(\xi)| \, d\xi$ is finite. More recently Ongie et al. (2019) used the *Radon transform* as a tool for analysis of infinite-width ReLU two-layer networks. In particular, they introduce the concept of Radon-based $\mathcal{R}$-norms and show that a function defined on $\mathbb{R}^d$ can be represented as an infinite-width two-layer neural network if and only if its $\mathcal{R}$-norm is finite. In this work, we extend the framework of (Ongie et al., 2019) and define similar Radon-based semi-norms $(\mathcal{R}, \mathcal{U}$-norms) such that a function admits an infinite-width neural network representation on a bounded open set $\mathcal{U} \subseteq \mathbb{R}^d$ when its $\mathcal{R}, \mathcal{U}$-norm is finite. Building on this, we derive sparse (finite-width) neural network approximation bounds that refine those of Breiman (1993); Klusowski & Barron (2018). Finally, we show that infinite-width neural network representations on bounded open sets are not unique and study their structure, providing a functional view of mode connectivity.

## 1 INTRODUCTION

Extensive work has shown that for a neural network to be able to generalize, the size or magnitude of the parameters is more important than the size of the network, when the latter is large enough (Bartlett, 1997; Neyshabur et al., 2015; Zhang et al., 2016). Under certain regimes, the size of the neural networks used in practice is so large that the training data is fit perfectly and an infinite-width approximation is appropriate. In this setting, what matters to obtain good generalization is to fit the data using the right inductive bias, which is specified by how network parameters are controlled (Wei et al., 2020) together with the training algorithm used (Lyu & Li, 2020).

The infinite-width two-layer neural network model has been studied from several perspectives due to its simplicity. One can replace the finite-width ReLU network $\frac{1}{n} \sum_{i=1}^{n} a_i (\langle \omega_i, x \rangle - b_i)_+$ by an integral over the parameter space with respect to a signed Radon measure: $\int (\langle \omega, x \rangle - b)_+ \, d\alpha(\omega, b)$. Thus, controlling the magnitude of the neural network parameters is akin to controlling the measure $\alpha$ according to a certain norm. Bach (2017) introduced the $\mathcal{F}_1$-space, which is the infinite-width neural network space with norm $\inf\{\int |b| \, d|\alpha|(\omega, b)\}$, derived from the finite-width regularizer $\frac{1}{n} \sum_{i=1}^{n} |a_i| \|(\omega_i, b_i)\|_2$ (the infimum is over all the measures $\alpha$ which represent the function at hand). A different line of work (Savarese et al., 2019; Ongie et al., 2019) consider the infinite-width spaces with norm $\inf\{\|\alpha\|_{\text{TV}} = \int d|\alpha|(\omega, b)\}$, which is derived from the finite-width regularizer $\frac{1}{n} \sum_{i=1}^{n} |a_i| \|\omega_i\|_2$ (i.e. omitting the bias term). Both of these works seek to find expressions for this norm, leading to characterizations of the functions that are representable by infinite-width networks. Savarese et al. (2019) solves the problem in the one-dimensional case: they show that for a function $f$ on $\mathbb{R}$, this norm takes value $\max\{\int_{\mathbb{R}} |f''(x)| \, dx, |f'(-\infty) + f'(\infty)|\}$. Ongie et al. (2019) give an expression for this norm (the $\mathcal{R}$-*norm*) for functions on $\mathbb{R}^d$, making use of Radon transforms (see Subsec. 2.3).

Although we mentioned in the first paragraph that in many occasions the network size is large enough that the specific number of neurons is irrelevant, when the target function is hard to approx-

imate it is interesting to have an idea of how many neurons one needs to approximate it. The first contribution in this direction was by Cybenko (1989); Hornik et al. (1989), which show that two-layer neural networks with enough neurons can approximate any reasonable function on bounded sets in the uniform convergence topology. Later on, Barron (1993); Breiman (1993) provided sparse approximation bounds stating that if a function $f$ is such that a certain quantity $C_f$ constructed from the Fourier transform $\hat{f}$ is finite, then there exists a neural network of width $n$ such that the $L^2$ approximation error with respect to a distribution of bounded support is lower than $O(C_f/n)$. More recently, Klusowski & Barron (2018) provided alternative sparse approximation bounds of Breiman (1993) by restricting to networks with bounded weights and a slightly better dependency on $n$ at the expense of a constant factor increasing with $d$ (see Subsec. 2.2).

**Contributions**. In our work, we seek to characterize the functions that coincide with an infinite-width two-layer neural network on a fixed bounded open set. This endeavor is interesting in itself because in practice, we want to learn target functions for which we know samples on a bounded set, and we are typically unconcerned with the values that the learned functions take at infinity. Moreover, the tools that we develop allow us to derive state-of-the-art sparse approximation bounds. Our main contributions are the following:

- In the spirit of the $\mathcal{R}$-norm introduced by Ongie et al. (2019), for any bounded open set $\mathcal{U} \subseteq \mathbb{R}^d$ we define the $\mathcal{R}, \mathcal{U}$-norm of a function on $\mathbb{R}^d$, and show that when the $\mathcal{R}, \mathcal{U}$-norm of $f$ is finite, $f(x)$ can admits a representation of the form $\int_{\mathbb{S}^{d-1} \times \mathbb{R}} (\langle \omega, x \rangle - b)_+ \, d\alpha(\omega, b) + \langle v, x \rangle + c$ for $x \in \mathcal{U}$, where $v \in \mathbb{R}^d$, $c \in \mathbb{R}$ and $\alpha$ is an even signed Radon measure.

- Using the $\mathcal{R}, \mathcal{U}$-norm, we derive function approximation bounds for neural networks with a fixed finite width. We compute the $\mathcal{R}, \mathcal{U}$-norm of a function in terms of its Fourier representation, and show that it admits an upper bound by the quantity $C_f$. This shows that our approximation bound is tighter than the previous bound by Breiman (1993), and meaningful in more instances (e.g. for finite-width neural networks). We also show $\mathcal{R}, \mathcal{U}$-norm-based bounds analogous to the ones of Klusowski & Barron (2018).

- Setting $\mathcal{U}$ as the open unit ball of radius $R$, we show that neural network representations of $f$ on $\mathcal{U}$ hold for multiple even Radon measures, which contrasts with the uniqueness result provided by Ongie et al. (2019) for the case of $\mathbb{R}^d$. We study the structure of the sets of Radon measures which give rise to the same function on $\mathcal{U}$. The non-uniqueness of the measure representing a measure could be linked to the phenomenon of mode connectivity.

**Additional related work**. There have been other recent works which have used the Radon transform to study neural networks in settings different from ours (Parhi & Nowak, 2021a; Bartolucci et al., 2021). These two works consider the $\mathcal{R}$-norm as a regularizer for an inverse problem, and proceed to prove representer theorems: there exists a solution of the regularized problem which is a two-layer neural network equal to the number of datapoints. Regarding infinite-width network spaces, E & Wojtowytsch (2020) present several equivalent definitions and provides a review. A well-known line of work (Mei et al., 2018; Chizat & Bach, 2018; Rotskoff & Vanden-Eijnden, 2018) studies the convergence of gradient descent for infinite-width two-layer neural networks.

## 2 FRAMEWORK

### 2.1 NOTATION

$\mathbb{S}^{d-1}$ denotes the $(d-1)$-dimensional hypersphere (as a submanifold of $\mathbb{R}^d$) and $\mathcal{B}_R(\mathbb{R}^d)$ is the Euclidean open ball of radius $R$. For $U \subseteq \mathbb{R}^d$ measurable, the space $C_0(U)$ of functions vanishing at infinity contains the continuous functions $f$ such that for any $\epsilon > 0$, there exists compact $K \subseteq U$ depending on $f$ such that $|f(x)| < \epsilon$ for $x \in U \setminus K$. $\mathcal{P}(U)$ is the set of Borel probability measures, $\mathcal{M}(U)$ is the space of finite signed Radon measures (which may be seen as the dual of $C_0(U)$). Throughout the paper, the term Radon measure refers to a finite signed Radon measure for shortness. If $\gamma \in \mathcal{M}(U)$, then $\|\gamma\|_{\mathrm{TV}}$ is the total variation (TV) norm of $\gamma$. $\mathcal{M}_{\mathbb{C}}(U)$ denotes the space of complex-valued finite signed Radon measures, defined as the dual space of $C_0(U, \mathbb{C})$ (the space of complex-valued functions vanishing at infinity). We denote by $\mathcal{S}(\mathbb{R}^d)$ the space of

Schwartz functions, which contains the functions in $\mathcal{C}^\infty(\mathbb{R}^d)$ whose derivatives of any order decay faster than polynomials of all orders, i.e. for all $k, p \in (\mathbb{N}_0)^d$, $\sup_{x \in \mathbb{R}^d} |x^k \partial^{(p)} \varphi(x)| < +\infty$. For $f \in L^1(\mathbb{R}^d)$, we use $\hat{f}$ to denote the unitary Fourier transforms with angular frequency, defined as $\hat{f}(\xi) = \frac{1}{(2\pi)^{d/2}} \int_{\mathbb{R}^d} f(x) e^{-i\langle \xi, x \rangle} dx$. If $\hat{f} \in L^1(\mathbb{R}^d)$ as well, we have the inversion formula $f(x) = \frac{1}{(2\pi)^{d/2}} \int_{\mathbb{R}^d} \hat{f}(\xi) e^{i\langle \xi, x \rangle} dx$. The Fourier transform is a continuous automorphism on $\mathcal{S}(\mathbb{R}^d)$.

## 2.2 Existing sparse approximation bounds

One of the classical results of the theory of two-layer neural networks (Breiman (1993), building on (Barron, 1993)) states that given a probability measure $p \in \mathcal{P}(\mathcal{B}_R(\mathbb{R}^d))$ and a function $f : \mathcal{B}_R(\mathbb{R}^d) \to \mathbb{R}$ admitting a Fourier representation of the form $f(x) = \frac{1}{(2\pi)^{d/2}} \int_{\mathbb{R}^d} e^{i\langle \xi, x \rangle} d\hat{f}(\xi)$, where $\hat{f} \in \mathcal{M}_\mathbb{C}(\mathbb{R}^d)$ is a complex-valued Radon measure such that $C_f = \frac{1}{(2\pi)^{d/2}} \int_{\mathbb{R}^d} \|\xi\|_2^2 \, d|\hat{f}|(\xi) < +\infty$, there exists a two-layer neural network $\tilde{f}(x) = \frac{1}{n} \sum_{i=1}^n a_i(\langle x, \omega_i \rangle - b_i)_+$ such that

$$\int_{\mathcal{B}_R(\mathbb{R}^d)} (f(x) - \tilde{f}(x))^2 \, dx \leq \frac{(2R)^4 C_f^2}{n}. \tag{1}$$

These classical results do not provide bounds on the magnitude of the neural network weights. More recently, Klusowski & Barron (2018) showed similar approximation bounds for two-layer ReLU networks under additional $l^1$ and $l^0$ bounds on the weights $a_i, \omega_i$. Namely, if $\tilde{C}_f = \frac{1}{(2\pi)^{d/2}} \int_{\mathbb{R}^d} \|\xi\|_1^2 \, d|\hat{f}|(\xi) < +\infty$ there exists a two-layer neural network $\tilde{f}(x) = a_0 + \langle \omega_0, x \rangle + \frac{\kappa}{n} \sum_{i=1}^n a_i(\langle w_i, x \rangle - b_i)_+$ with $|a_i| \leq 1$, $\|\omega_i\| \leq 1$, $b_i \in [0, 1]$, and $\kappa \leq 2\tilde{C}_f$, and

$$\sup_{x \in [-1,1]^d} |f(x) - \tilde{f}(x)| \leq c \, \tilde{C}_f \sqrt{d + \log n} \, n^{-1/2 - 1/d}, \tag{2}$$

where $c$ is a universal constant.

## 2.3 Representation results on $\mathbb{R}^d$ based on the Radon transform

One defines $\mathbb{P}^d$ denotes the space of hyperplanes on $\mathbb{R}^d$, whose elements may be represented by points in $\mathbb{S}^{d-1} \times \mathbb{R}$ by identifying $\{x | \langle \omega, x \rangle = b\}$ with both $(\omega, b)$ and $(-\omega, -b)$. Thus, functions on $\mathbb{P}^d$ are even functions on $\mathbb{S}^{d-1} \times \mathbb{R}$ and we will use both notions interchangeably[1].

**The Radon transform and the dual Radon transform.** If $f : \mathbb{R}^d \to \mathbb{R}$ is a function which is integrable over all the hyperplanes of $\mathbb{R}^d$, we may define the Radon transform $\mathcal{R}f : \mathbb{P}^d \to \mathbb{R}$ as

$$\mathcal{R}f(\omega, b) = \int_{\{x | \langle \omega, x \rangle = b\}} f(x) \, dx, \quad \forall (\omega, b) \in \mathbb{S}^{d-1} \times \mathbb{R}.$$

That is, one integrates the function $f$ over the hyperplane $(\omega, b)$. If $\Phi : \mathbb{P}^d \to \mathbb{R}$ is a continuous function, the dual Radon transform $\mathcal{R}^* \Phi : \mathbb{R}^d \to \mathbb{R}$ is defined as

$$\mathcal{R}^* \Phi(x) = \int_{\mathbb{S}^{d-1}} \Phi(\omega, \langle \omega, x \rangle) \, d\omega, \quad \forall x \in \mathbb{R}^d,$$

where the integral is with respect to the Hausdorff measure over $\mathbb{S}^{d-1}$. $\mathcal{R}$ and $\mathcal{R}^*$ are adjoint operators in the appropriate domains (see Lemma 13).

**The Radon inversion formula.** When $f \in C^\infty(\mathbb{R}^d)$, one has that (Theorem 3.1, Helgason (2011))

$$f = c_d(-\Delta)^{(d-1)/2} \mathcal{R}^* \mathcal{R} f \tag{3}$$

where $c_d = \frac{1}{2(2\pi)^{d-1}}$ and $(-\Delta)^{s/2}$ denotes the (negative) fractional Laplacian, defined via its Fourier transform as $\widehat{(-\Delta)^{s/2} f}(\xi) = \|\xi\|^s \hat{f}(\xi)$.

---

[1]Similarly, the space $\mathcal{M}(\mathbb{P}^d)$ of Radon measures over $\mathbb{P}^d$ contains the even measures in $\mathcal{M}(\mathbb{S}^{d-1} \times \mathbb{R})$. If $\alpha \in \mathcal{M}(\mathbb{P}^d)$, $\int_{\mathbb{S}^{d-1} \times \mathbb{R}} \varphi(\omega, b) \, d\alpha(\omega, b)$ is well defined for any measurable function $\varphi$ on $\mathbb{S}^{d-1} \times \mathbb{R}$, but $\int_{\mathbb{P}^d} \varphi(\omega, b) \, d\alpha(\omega, b)$ is only defined for even $\varphi$.

**The $\mathcal{R}$-norm.** Given a function $f : \mathbb{R}^d \to \mathbb{R}$, Ongie et al. (2019) introduce the quantity

$$\|f\|_{\mathcal{R}} = \begin{cases} \sup\{-c_d\langle f, (-\Delta)^{(d+1)/2}\mathcal{R}^*\psi\rangle \mid \psi \in \mathcal{S}(\mathbb{S}^{d-1} \times \mathbb{R}), \ \psi \text{ even}, \ \|\psi\|_\infty \leq 1\} & \text{if } f \text{ Lipschitz} \\ +\infty & \text{otherwise.} \end{cases}$$
(4)

They call it the $\mathcal{R}$-norm of $f$, although it is formally a semi-norm. Here, the space $\mathcal{S}(\mathbb{S}^{d-1} \times \mathbb{R})$ of Schwartz functions on $\mathbb{S}^{d-1} \times \mathbb{R}$ is defined, in analogy with $\mathcal{S}(\mathbb{R}^d)$, as the space of $C^\infty$ functions $\psi$ on $\mathbb{S}^{d-1} \times \mathbb{R}$ which for any integers $k, l \geq 0$ and any differential operator $D$ on $\mathbb{S}^{d-1}$ satisfy $\sup_{(\omega,b)\in\mathbb{S}^{d-1}\times\mathbb{R}} |(1 + |b|^k)\partial_b^k(D\psi)(\omega, b)| < +\infty$ (Helgason (2011), p. 5). Moreover, $\mathcal{S}(\mathbb{P}^d) = \{\psi \in \mathcal{S}(\mathbb{S}^{d-1} \times \mathbb{R}) \mid \psi \text{ even}\}$, which means the conditions on $\psi$ in (4) can be written as $\psi \in \mathcal{S}(\mathbb{P}^d), \|\psi\|_\infty \leq 1$.

The finiteness of the $\mathcal{R}$-norm indicates whether a function on $\mathbb{R}^d$ admits an exact representation as an infinitely wide neural network. Namely, Ongie et al. (2019) in their Lemma 10 show that $\|f\|_{\mathcal{R}}$ is finite if and only if there exists a (unique) even measure $\alpha \in \mathcal{M}(\mathbb{S}^{d-1} \times \mathbb{R})$ and (unique) $v \in \mathbb{R}^d$, $c \in \mathbb{R}$ such that for any $x \in \mathbb{R}^d$,

$$f(x) = \int_{\mathbb{S}^{d-1}\times\mathbb{R}} \big(\langle\omega, x\rangle - b\big)_+ \, d\alpha(\omega, b) + \langle v, x\rangle + c,$$
(5)

in which case, $\|f\|_{\mathcal{R}} = \|\alpha\|_{\mathrm{TV}}$.

Remark the following differences between this result and the bounds by (Breiman, 1993; Klusowski & Barron, 2018) shown in equations (1) and (2):

(i) in (5) we have an exact representation with infinite-width neural networks instead of an approximation result with finite-width,

(ii) in (5) the representation holds on $\mathbb{R}^d$ instead of a bounded domain.

In our work, we derive representation results similar to the ones of Ongie et al. (2019) for functions defined on bounded open sets, which naturally give rise to sparse approximation results that refine those of (Breiman, 1993; Klusowski & Barron, 2018).

One property that makes the Radon transform and its dual useful to analyze neural networks can be understood at a very high level via the following argument: if $f(x) = \int_{\mathbb{S}^{d-1}\times\mathbb{R}}(\langle\omega, x\rangle - b)_+\rho(\omega, b) \, d(\omega, b) + \langle v, x\rangle + c$ for some smooth rapidly decreasing function $\rho$, then $\Delta f(x) = \int_{\mathbb{S}^{d-1}\times\mathbb{R}} \delta_{\langle\omega,x\rangle=b}\rho(\omega, b) \, d(\omega, b) = \int_{\mathbb{S}^{d-1}} \rho(\omega, \langle\omega, x\rangle) \, d\omega = (\mathcal{R}^*\rho)(x)$. For a general function $f$ of the form (5), one has similarly that $\langle\Delta f, \varphi\rangle = \langle\alpha, \mathcal{R}\varphi\rangle$ for any $\varphi \in \mathcal{S}(\mathbb{R}^d)$. This property relates the evaluations of the measure $\alpha$ to the function $\Delta f$ via the Radon transform, and is the main ingredient in the proof of Lemma 10 of Ongie et al. (2019). While we also rely on it, we need many additional tools to deal with the case of bounded open sets.

## 3 REPRESENTATION RESULTS ON BOUNDED OPEN SETS

**Schwartz functions on open sets.** Let $\mathcal{U} \subseteq \mathbb{R}^d$ be an open subset. The space of Schwartz functions on $\mathcal{U}$ may be defined as $\mathcal{S}(\mathcal{U}) = \bigcap_{z\in\mathbb{R}^d\setminus\mathcal{U}} \bigcap_{k\in(\mathbb{N}_0)^d}\{f \in \mathcal{S}(\mathbb{R}^d) \mid \partial^{(k)}f(z) = 0\}$, i.e. they are those Schwartz functions on $\mathbb{R}^d$ such that the derivatives of all orders vanish outside of $\mathcal{U}$ (c.f. Def. 3.2, Shaviv (2020)). The structure of $\mathcal{S}(\mathcal{U})$ is similar to $\mathcal{S}(\mathbb{R}^d)$ in that its topology is given by a family of semi-norms indexed by $((\mathbb{N}_0)^d)^2$: $\|f\|_{k,k'} = \sup_{x\in\mathcal{U}} |x^k \cdot f^{(k')}(x)|$. Similarly, if $\mathcal{V} \subseteq \mathbb{P}^d$ is open, we define $\mathcal{S}(\mathcal{V}) = \bigcap_{(\omega,b)\in(\mathbb{S}^{d-1}\times\mathbb{R})\setminus\mathcal{V}} \bigcap_{k\in(\mathbb{N}_0)^2}\{f \in \mathcal{S}(\mathbb{P}^d) \mid \partial_b^{k_1}\hat{\Delta}^{k_2}f(\omega, b) = 0\}$, where $\hat{\Delta}$ is the spherical Laplacian.

**The $\mathcal{R}, \mathcal{U}$-norm.** Let $\mathcal{U} \subseteq \mathbb{R}^d$ be a bounded open set, and let $\tilde{\mathcal{U}} := \{(\omega, \langle\omega, x\rangle) \in \mathbb{S}^{d-1} \times \mathbb{R} \mid x \in \mathcal{U}\}$. For any function $f : \mathbb{R}^d \to \mathbb{R}$, we define the $\mathcal{R}, \mathcal{U}$-norm of $f$ as

$$\|f\|_{\mathcal{R},\mathcal{U}} = \sup\{-c_d\langle f, (-\Delta)^{(d+1)/2}\mathcal{R}^*\psi\rangle \mid \psi \in \mathcal{S}(\tilde{\mathcal{U}}), \ \psi \text{ even}, \ \|\psi\|_\infty \leq 1\}.$$
(6)

Note the similarity between this quantity and the $\mathcal{R}$-norm defined in (4); the main differences are that the supremum here is taken over the even Schwartz functions on $\tilde{\mathcal{U}}$ instead of $\mathbb{S}^{d-1} \times \mathbb{R}$, and that the non-Lipschitz case does not need a separate treatment. Remark that $\|f\|_{\mathcal{R},\mathcal{U}} \leq \|f\|_{\mathcal{R}}$. If $f$ has enough regularity, we may write $\|f\|_{\mathcal{R},\mathcal{U}} = \int_{\tilde{\mathcal{U}}} |\mathcal{R}(-\Delta)^{(d+1)/2}f|(\omega, b)\, d(\omega, b)$, using that the fractional Laplacian is self-adjoint and $\mathcal{R}^*$ is the adjoint of $\mathcal{R}$.

Define $\mathbb{P}_{\mathcal{U}}^d$ to be the bounded open set of hyperplanes of $\mathbb{R}^d$ that intersect $\mathcal{U}$, which in analogy with Subsec. 2.3, is equal to $\tilde{\mathcal{U}}$ up to the identification of $(\omega, b)$ with $(-\omega, -b)$. Similarly, note that $\mathcal{S}(\mathbb{P}_{\mathcal{U}}^d) = \{\psi \in \mathcal{S}(\tilde{\mathcal{U}}), \psi \text{ even}\}$, which allows to rewrite the conditions in (6) as $\psi \in \mathcal{S}(\mathbb{P}_{\mathcal{U}}^d), \|\psi\|_\infty \leq 1$.

The following proposition, which is based on the Riesz-Markov-Kakutani representation theorem, shows that when the $\mathcal{R},\mathcal{U}$-norm is finite, it can be associated to a unique Radon measure over $\mathbb{P}_{\mathcal{U}}^d$.

**Proposition 1.** *If $\|f\|_{\mathcal{R},\mathcal{U}} < +\infty$, there exists a unique Radon measure $\alpha \in \mathcal{M}(\mathbb{P}_{\mathcal{U}}^d)$ such that $-c_d\langle f, (-\Delta)^{(d+1)/2}\mathcal{R}^*\psi\rangle = \int_{\mathbb{P}_{\mathcal{U}}^d} \psi(\omega, b)\, d\alpha(\omega, b)$ for any $\psi \in \mathcal{S}(\mathbb{P}_{\mathcal{U}}^d)$. Moreover, $\|f\|_{\mathcal{R},\mathcal{U}} = \|\alpha\|_{TV}$.*

Building on this, we see that a neural network representation for bounded $\mathcal{U}$ holds when the $\mathcal{R},\mathcal{U}$-norm is finite:

**Theorem 1.** *Let $\mathcal{U}$ be a open, bounded subset of $\mathbb{R}^d$. Let $f : \mathbb{R}^d \to \mathbb{R}$ such that $\|f\|_{\mathcal{R},\mathcal{U}} < +\infty$. Let $\alpha \in \mathcal{M}(\mathbb{P}_{\mathcal{U}}^d)$ be given by Proposition 1. For any $\varphi \in \mathcal{S}(\mathcal{U})$, there exist unique $v \in \mathbb{R}^d$ and $c \in \mathbb{R}$ such that*

$$\int_{\mathcal{U}} f(x)\varphi(x)\, dx = \int_{\mathcal{U}} \left( \int_{\tilde{\mathcal{U}}} (\langle \omega, x \rangle - t)_+ \, d\alpha(\omega, t) + \langle v, x \rangle + c \right) \varphi(x)\, dx, \tag{7}$$

*That is, $f(x) = \int_{\tilde{\mathcal{U}}} (\langle \omega, x \rangle - t)_+ \, d\alpha(\omega, t) + \langle v, x \rangle + c$ for $x$ a.e. (almost everywhere) in $\mathcal{U}$. If $f$ is continuous, then the equality holds for all $x \in \mathcal{U}$.*

Remark that this theorem does not claim that the representation given by $\alpha, v, c$ is unique, unlike Lemma 10 by Ongie et al. (2019) concerning analogous representations on $\mathbb{R}^d$. In Sec. 5 we see that such representations are in fact not unique, for particular choices of the set $\mathcal{U}$. We want to underline that the proof of Theorem 1 uses completely different tools from the ones of Lemma 10 by Ongie et al. (2019): their result relies critically on the fact that the only harmonic Lipschitz functions on $\mathbb{R}^d$ are affine functions, which is not true for functions on bounded subsets in our setting.

## 4 Sparse approximation for functions with bounded $\mathcal{R},\mathcal{U}$-norm

In this section, we show how to obtain approximation bounds of a function $f$ on a bounded open set $\mathcal{U}$ using a fixed-width neural network with bounded coefficients, in terms of the $\mathcal{R},\mathcal{U}$-norm introduced in the previous section.

**Theorem 2.** *Let $\mathcal{U} \subseteq \mathcal{B}_R(\mathbb{R}^d)$ be a bounded open set. Suppose that $f : \mathbb{R}^d \to \mathbb{R}$ is such that $\|f\|_{\mathcal{R},\mathcal{U}}$ is finite, where $\|\cdot\|_{\mathcal{R},\mathcal{U}}$ is defined in (6). Let $v \in \mathbb{R}^d, c \in \mathbb{R}$ as in Theorem 1. Then, there exists $\{(\omega_i, b_i)\}_{i=1}^n \subseteq \tilde{\mathcal{U}}$ and $\{a_i\}_{i=1}^n \subseteq \{\pm 1\}$ such that the function $\tilde{f} : \mathbb{R}^d \to \mathbb{R}$ defined as*

$$\tilde{f}(x) = \frac{\|f\|_{\mathcal{R},\mathcal{U}}}{n} \sum_{i=1}^n a_i(\langle \omega_i, x \rangle - b_i)_+ + \langle v, x \rangle + c$$

*fulfills, for $x$ a.e. in $\mathcal{U}$,*

$$\left| \tilde{f}(x) - f(x) \right| \leq \frac{R\|f\|_{\mathcal{R},\mathcal{U}}}{\sqrt{n}}. \tag{8}$$

*The equality holds for all $x \in \mathcal{U}$ if $f$ is continuous.*

The proof of Theorem 2 (in App. B) uses the neural network representation (7) and a probabilistic argument. If one samples $\{(\omega_i, b_i)\}_{i=1}^n$ from a probability distribution proportional to $|\alpha|$, a

Rademacher complexity bound upper-bounds the expectation of the supremum norm between $\tilde{f}$ and $f$, which yields the result.

Note the resemblance of (8) with the bound (1); the $\mathcal{R}, \mathcal{U}$ norm of $f$ replaces the quantity $C_f$. We can also use the $\mathcal{R}, \mathcal{U}$-norm to obtain a bound analogous to (2), that is, with a slightly better dependency in the exponent of $n$ at the expense of a constant factor growing with the dimension.

**Proposition 2.** *Let $f : \mathbb{R}^d \to \mathbb{R}$ and $\mathcal{U} \subseteq \mathcal{B}_1(\mathbb{R}^d)$ open such that $\|f\|_{\mathcal{R}, \mathcal{U}} < +\infty$. Then, then there exist $\{a_i\}_{i=1}^n \subseteq [-1, 1]$, $\{\omega_i\}_{i=1}^n \subseteq \{\omega \in \mathbb{R}^d | \|\omega\|_1 = 1\}$ and $\{b_i\}_{i=1}^n \subseteq [0, 1]$ and $\kappa < \sqrt{d}\|f\|_{\mathcal{R}, \mathcal{U}}$ such that the function*

$$\tilde{f}(x) = \frac{\kappa}{n} \sum_{i=1}^n a_i (\langle \omega_i, x \rangle - b_i)_+$$

*fulfills, for $x$ a.e. in $\mathcal{U}$ and some universal constant $c > 0$,*

$$|f(x) - \tilde{f}(x)| \leq c\kappa \sqrt{d + \log n}\, n^{-1/2 - 1/d}.$$

The proof of this result (in App. B) follows readily from the representation (7) and Theorem 1 of Klusowski & Barron (2018).

### 4.1 LINKS WITH THE FOURIER SPARSE APPROXIMATION BOUNDS

The following result shows that setting $\mathcal{U} = \mathcal{B}_R(\mathbb{R}^d)$, the $\mathcal{R}, \mathcal{U}$-norm can be bounded by the Fourier-based quantities $C_f, \tilde{C}_f$ introduced in Subsec. 2.2.

**Theorem 3.** *Assume that the function $f : \mathbb{R}^d \to \mathbb{R}$ admits a Fourier representation of the form $f(x) = \frac{1}{(2\pi)^{d/2}} \int_{\mathbb{R}^d} e^{i\langle \xi, x \rangle} d\hat{f}(\xi)$ with $\hat{f} \in \mathcal{M}_\mathbb{C}(\mathbb{R}^d)$ a complex-valued Radon measure. Let $C_f$ be the quantity used in the sparse approximation bound by Breiman (1993) (see Subsec. 2.2). Then, one has that*

$$\|f\|_{\mathcal{R}, \mathcal{B}_R(\mathbb{R}^d)} \leq 2R C_f \tag{9}$$

As a direct consequence of Theorem 3, when $\mathcal{U} = \mathcal{B}_R(\mathbb{R}^d)$ the right-hand side of (8) can be upper-bounded by $R^2 C_f / \sqrt{n}$. This allows to refine the bound (1) from Breiman (1993) to a bound in the supremum norm over $\mathcal{B}_R(\mathbb{R}^d)$, and where the approximating neural network $\tilde{f}(x) = \frac{1}{n} \sum_{i=1}^n a_i (\langle x, \omega_i \rangle - b_i)_+ + \langle v, x \rangle + c$ fulfills $|a_i| \leq \|f\|_{\mathcal{R}, \mathcal{B}_R(\mathbb{R}^d)}$, $\|\omega_i\|_2 \leq 1$ and $b_i \in (-R, R)$.

While we are able to prove the bound (9), the Fourier representation of $f$ does not allow for a manageable expression for the measure $\alpha$ described in Proposition 1. For that, the following theorem starts from a slightly modified Fourier representation of $f$, under which one can describe the measure $\alpha$ and provide a formula for the $\mathcal{R}, \mathcal{U}$-norm.

**Theorem 4.** *Let $f : \mathbb{R}^d \to \mathbb{R}$ admitting the representation*

$$f(x) = \int_{\mathbb{S}^{d-1} \times \mathbb{R}} e^{ib\langle \omega, x \rangle} d\mu(\omega, b), \tag{10}$$

*for some complex-valued Radon measures $\mu \in \mathcal{M}_\mathbb{C}(\mathbb{S}^{d-1} \times \mathbb{R})$ such that $d\mu(\omega, b) = d\mu(-\omega, -b) = d\bar{\mu}(-\omega, b) = d\bar{\mu}(\omega, -b)$, and $\int_{\mathbb{S}^{d-1} \times \mathbb{R}} b^2 d|\mu|(\omega, b) < +\infty$. Choosing $\mathcal{U} = \mathcal{B}_R(\mathbb{R}^d)$, the unique measure $\alpha \in \mathcal{M}(\mathbb{P}_R^d)$ specified by Proposition 1 takes the following form:*

$$d\alpha(\omega, b) = -\int_\mathbb{R} t^2 e^{-itb} d\mu(\omega, t)\, db,$$

*where $K = 2\pi^{d/2}/\Gamma(\frac{d}{2})$. Note that $\alpha$ is real-valued because $\int_\mathbb{R} t^2 e^{-itb} d\mu(\omega, t) \in \mathbb{R}$ as $\overline{t^2 d\mu(\omega, t)} = (-t)^2 d\mu(\omega, -t)$. Consequently, the $\mathcal{R}, \mathcal{B}_R(\mathbb{R}^d)$-norm of $f$ is*

$$\|f\|_{\mathcal{R}, \mathcal{B}_R(\mathbb{R}^d)} = \|\alpha\|_{TV} = \int_{-R}^R \int_{\mathbb{S}^{d-1}} \left| \int_\mathbb{R} t^2 e^{-itb} d\mu(\omega, t) \right| db. \tag{11}$$

Remark that $\mu$ is the pushforward of the measure $\hat{f}$ by the mappings $\xi \mapsto (\pm\xi/\|\xi\|, \pm\xi)$. When the Fourier transform $\hat{f}$ admits a density, one may obtain the density of $\mu$ via a change from Euclidean to spherical coordinates: $d\mu(\omega, b) = \frac{1}{2}\text{vol}(\mathbb{S}^{d-1})\hat{f}(b\omega)|b|^{d-1} d(\omega, b)$. Hence, Theorem 4 provides an operative way to compute the $\mathcal{R}, \mathcal{U}$-norm of $f$ if one has access to the Fourier transform of $\hat{f}$. Note that equation (11) implies that the $\mathcal{R}, \mathcal{B}_R(\mathbb{R}^d)$-norm of $f$ increases with $R$, and in particular is smaller than the $\mathcal{R}$-norm of $f$, which corresponds to setting $R = \infty$.

Theorems 3 and 4 are proven jointly in App. B. Note that from the expression (11) one can easily see that $\|f\|_{\mathcal{R}, \mathcal{B}_R(\mathbb{R}^d)}$ is upper-bounded by $RC_f$:

$$\int_{-R}^{R} \int_{\mathbb{S}^{d-1}} \left| \int_{\mathbb{R}} t^2 e^{-itb} d\mu(\omega, t) \right| db \leq \int_{-R}^{R} \int_{\mathbb{S}^{d-1}} \int_{\mathbb{R}} t^2 d|\mu|(\omega, t) db = 2R \int_{\mathbb{R}^d} \|\xi\|^2 d|\hat{f}|(\xi) \quad (12)$$

where the equality holds since $\mu$ is the pushforward of $\hat{f}$. Equation (12) makes apparent the norm $\|f\|_{\mathcal{R}, \mathcal{B}_R(\mathbb{R}^d)}$ is sometimes much smaller than the quantities $C_f, \tilde{C}_f$, as is showcased by the following one-dimensional example (see the proof in App. B). In these situations, the sparse approximation bounds that we provide in Theorem 2 and Proposition 2 are much better than the ones in (1)-(2).

**Example 1.** *Take the function $f : \mathbb{R} \rightarrow \mathbb{R}$ defined as $f(x) = \cos(x) - \cos((1 + \epsilon)x)$, with $\epsilon > 0$. $f$ admits the Fourier representation $f(x) = \frac{1}{(2\pi)^{1/2}} \int_{\mathbb{R}} \sqrt{\frac{\pi}{2}}(\delta_1(\xi) + \delta_{-1}(\xi) - \delta_{1+\epsilon}(\xi) - \delta_{-1-\epsilon}(\xi))e^{i\xi x} d\xi$. We have that $C_f = 2 + 2\epsilon + \epsilon^2$, and $\|f\|_{\mathcal{R}, \mathcal{B}_R(\mathbb{R}^d)} \leq R(R\epsilon + 2\epsilon + \epsilon^2)$. $\|f\|_{\mathcal{R}, \mathcal{B}_R(\mathbb{R}^d)}$ goes to zero as $\epsilon \rightarrow 0$, while $C_f$ converges to 2.*

An interesting class of functions for which $\|f\|_{\mathcal{R}, \mathcal{B}_R(\mathbb{R}^d)}$ is finite but $C_f, \tilde{C}_f$ are infinite are functions that can be written as a finite-width neural network on $\mathcal{B}_R(\mathbb{R}^d)$, as shown in the following proposition.

**Proposition 3.** *Let $f : \mathbb{R}^d \rightarrow \mathbb{R}$ defined as $f(x) = \frac{1}{n} \sum_{i=1}^{n} a_i(\langle\omega_i, x\rangle - b_i)_+$ for all $x \in \mathbb{R}^d$, with $\{\omega_i\}_{i=1}^{n} \subseteq \mathbb{S}^{d-1}$, $\{a_i\}_{i=1}^{n}, \{b_i\}_{i=1}^{n} \subseteq \mathbb{R}$. Then, for any bounded open set $\mathcal{U}$, we have $\|f\|_{\mathcal{R}, \mathcal{U}} \leq \frac{1}{n}\sum_{i=1}^{n}|a_i|$, while $C_f, \tilde{C}_f = +\infty$ if $f$ is not an affine function.*

Proposition 3 makes use of the fact that the $\mathcal{R}, \mathcal{U}$-norm is always upper-bounded by the $\mathcal{R}$-norm, which also means that all the bounds developed in Ongie et al. (2019) apply for the $\mathcal{R}, \mathcal{U}$-norm. The fact that finite-width neural networks have infinite $C_f$ was stated by E & Wojtowytsch (2020), that used them to show the gap between the functions with finite $C_f$ and functions representable by infinite-width neural networks (belonging to the Barron space, in their terminology). It remains to be seen whether the gap is closed when considering functions with finite $\mathcal{R}, \mathcal{U}$-norm, i.e., whether any function admitting an infinite-width representation (7) on $\mathcal{U}$ has a finite $\mathcal{R}, \mathcal{U}$-norm.

**Moving to the non-linear Radon transform.** In many applications the function of interest $f$ may be better represented as $\int(\langle\omega, \varphi(x)\rangle - t)_+ d\alpha(\omega, t) + \langle v, x\rangle + c$, where $\varphi$ is a fixed finite dimensional, non-linear and bounded feature map. Our results trivially extend to this case where in the Radon transform hyperplanes are replaced by hyperplanes in the feature space. This can be seen as the "kernel trick" applied to the Radon transform. The corresponding $\|f\|_{\mathcal{R}, \varphi(\mathcal{U})}$ corresponds to the sparsity of the decomposition in the feature space, and we have better approximation when $\|f\|_{\mathcal{R}, \varphi(\mathcal{U})} < \|f\|_{\mathcal{R}, \mathcal{U}}$. This gives a simple condition for when transfer learning is successful, and explains the success of using random fourier features as a preprocessing in implicit neural representations of images and surfaces (Tancik et al., 2020). In order to go beyond the fixed feature maps and tackle deeper ReLU networks, we think that the non-linear Radon transform (Ehrenpreis, 2003) is an interesting tool to explore. We note that Parhi & Nowak (2021b) introduced recently a representer theorem for deep ReLU networks using Radon transforms as a regularizer.

## 5 INFINITE-WIDTH REPRESENTATIONS ARE NOT UNIQUE ON BOUNDED SETS

Ongie et al. (2019) show that when the $\mathcal{R}$-norm of $f$ is finite, there is a unique measure $\alpha \in \mathcal{M}(\mathbb{R}^d)$ such that the representation (5) holds for $x \in \mathbb{R}^d$. In this section we show that when we only ask

the representation to hold for $x$ in a bounded open set, there exist several measures that do the job; in fact, they span an infinite-dimensional space.

Let $\mathcal{U} = \mathcal{B}_R(\mathbb{R}^d)$ be the open ball of radius $R > 0$ in $\mathbb{R}^d$, which means that $\tilde{\mathcal{U}} = \mathbb{S}^{d-1} \times (-R, R)$ and $\mathbb{P}_{\mathcal{U}}^d$ is the set of hyperplanes $\{x | \langle \omega, x \rangle = b\}$ such that $\|\omega\| = 1$ and $b \in (-R, R)$, which we denote by $\mathbb{P}_R^d$ for simplicity. In the following we will construct a space of Radon measures $\alpha \in \mathcal{M}(\mathbb{P}_R^d)$ whose neural network representation (5) coincide for all $x \in \mathcal{B}_R(\mathbb{R}^d)$. Note that since any bounded subset of $\mathbb{R}^d$ is included in some open ball, our results imply that such representations are non-unique on any bounded set.

**Remark 1.** *When one considers representations on $\mathcal{B}_R(\mathbb{R}^d)$ of the sort (5) with the measure $\alpha$ lying in the larger space $\mathcal{M}(\mathbb{S}^{d-1} \times \mathbb{R})$, the non-uniqueness is apparent because there are two 'trivial' kinds of symmetry at play:*

*(i) Related to parity: when the measure $\alpha$ is odd, we have $\int_{\mathbb{S}^{d-1} \times \mathbb{R}} (\langle \omega, x \rangle - b)_+ \, d\alpha(\omega, b) = \frac{1}{2} \int_{\mathbb{S}^{d-1} \times \mathbb{R}} (\langle \omega, x \rangle - b)_+ - (-\langle \omega, x \rangle + b)_+ \, d\alpha(\omega, b) = \langle \frac{1}{2} \int_{\mathbb{S}^{d-1} \times \mathbb{R}} \omega \, d\alpha(\omega, b), x \rangle - \frac{1}{2} \int_{\mathbb{S}^{d-1} \times \mathbb{R}} b \, d\alpha(\omega, b)$, which is an affine function of $x$.*

*(ii) Related to boundedness: if $(\omega, b) \in \mathbb{S}^{d-1} \times (\mathbb{R} \setminus (-R, R))$, $x \mapsto (\langle \omega, x \rangle - b)_+$ restricted to $\mathcal{B}_R(\mathbb{R}^d)$ is an affine function of $x$. Hence, if $\alpha$ is supported on $\mathbb{S}^{d-1} \times \mathbb{S}^{d-1} \times (\mathbb{R} \setminus (-R, R))$, $x \mapsto \int_{\mathbb{S}^{d-1} \times \mathbb{R}} (\langle \omega, x \rangle - b)_+ \, d\alpha(\omega, b)$ is an affine function when restricted to $\mathcal{B}_R(\mathbb{R}^d)$.*

*Since in Sec. 3 we restrict our scope to measures $\alpha$ lying in $\mathcal{M}(\mathbb{P}_{\mathcal{U}}^d)$, these two kinds of symmetries are already quotiented out in our analysis. The third kind of non-uniqueness that we discuss in this section is conceptually deeper, taking place within $\mathcal{M}(\mathbb{P}_{\mathcal{U}}^d)$.*

Let $\{Y_{k,j} \mid k \in \mathbb{Z}^+, \ 1 \leq j \leq N_{k,d}\}$ be the orthonormal basis of spherical harmonics of the space $L^2(\mathbb{S}^{d-1})$ (Atkinson & Han, 2012). It is well known that for any $k$, the functions $\{Y_{k,j} \mid 1 \leq j \leq N_{k,d}\}$ are the restrictions to $\mathbb{S}^{d-1}$ of homogeneous polynomials of degree $k$, and in fact $N_{k,d}$ is the dimension of the space of homogeneous harmonic polynomials of degree $k$. Consider the following subset of even functions in $C^\infty(\mathbb{S}^{d-1} \times (-R, R))$:

$$A = \{Y_{k,j} \otimes X^{k'} \mid k, j, k' \in \mathbb{Z}^+, \ k \equiv k' \pmod{2}, \ k' < k - 2, \ 1 \leq j \leq N_{d,k}\},$$

where $X^{k'}$ denotes the monomial of degree $k'$ on $(-R, R)$. We have the following result regarding the non-uniqueness of neural network representations:

**Theorem 5.** *If $\alpha \in \mathcal{M}(\mathbb{P}_R^d)$ is such that $\alpha \in cl_w(span(A))$, then we have that $0 = \int_{\mathbb{S}^{d-1} \times (-R, R)} (\langle \omega, x \rangle - b)_+ \, d\alpha(\omega, b)$ for any $x \in \mathcal{B}_R(\mathbb{R}^d)$. That is, $\alpha$ yields a neural network representation of the zero-function on $\mathcal{B}_R(\mathbb{R}^d)$. Here, we consider $span(A)$ as a subset of $\mathcal{M}(\mathbb{P}_R^d)$ by the Riesz-Markov-Kakutani representation theorem via the action $\langle g, \varphi \rangle = \int_{\mathbb{P}_R^d} \varphi(\omega, b) g(\omega, b) \, d(\omega, b)$ for any $g \in span(A), \varphi \in C_0(\mathbb{P}_R^d)$, and $cl_w$ denotes the closure in the topology of weak convergence of $\mathcal{M}(\mathbb{S}^{d-1} \times \mathbb{R})$.*

In particular, any measure whose density is in the span of $A$ will yield a function which is equal to zero when restricted to $\mathcal{B}_R(\mathbb{R}^d)$. As an example of this result, we show a simple measure in $\mathcal{M}(\mathbb{P}_1^d)$ which represents the zero function on $\mathcal{B}_1(\mathbb{R}^2)$.

**Example 2** (Non-zero measure representing the zero function on $\mathcal{B}_1(\mathbb{R}^2)$). *We define the even Radon measure $\alpha \in \mathcal{M}(\mathbb{S}^1 \times (-1, 1))$ with density $d\alpha(\omega, b) = (8\omega_0^4 - 8\omega_0^2 + 1) \, d(\omega, b)$ where $\omega = (\omega_0, \omega_1)$. Then, for any $x \in \mathcal{B}_1(\mathbb{R}^2)$, $0 = \int_{\mathbb{S}^1 \times (-1, 1)} (\langle \omega, x \rangle - b)_+ \, d\alpha(\omega, x)$.*

On the one hand, Proposition 1 states that there exists a unique measure $\alpha \in \mathcal{M}(\mathbb{P}_{\mathcal{U}}^d)$ such that $-c_d \langle f, (-\Delta)^{(d+1)/2} \mathcal{R}^* \psi \rangle = \int_{\mathbb{P}_{\mathcal{U}}^d} \psi(\omega, b) \, d\alpha(\omega, b)$ for any $\psi \in \mathcal{S}(\mathbb{P}_{\mathcal{U}}^d)$ if $\|f\|_{\mathcal{R}, \mathcal{U}}$ is finite. On the other hand, Theorem 5 claims that functions admit distinct representations by measures in $\mathcal{M}(\mathbb{P}_{\mathcal{U}}^d)$. The following theorem clarifies these two seemingly contradictory statements. Consider the following subset of even functions in $C^\infty(\mathbb{S}^{d-1} \times (-R, R))$, which contains $A$:

$$B = \{Y_{k,j} \otimes X^{k'} \mid k, j, k' \in \mathbb{Z}^+, \ k \equiv k' \pmod{2}, \ k' < k, \ 1 \leq j \leq N_{d,k}\}.$$

**Proposition 4.** *Let $0 < R < R'$. Let $f : \mathbb{R}^d \to \mathbb{R}$ such that $\|f\|_{\mathcal{R}, \mathcal{B}_{R'}(\mathbb{R}^d)} < +\infty$ and let $\alpha \in \mathcal{M}(\mathbb{P}_R^d)$ be the unique measure specified by* Proposition 1. *Then, $\alpha$ is the unique measure in $\mathcal{M}(\mathbb{P}_R^d)$ such that*

$$\forall \varphi \in \mathcal{S}(\mathcal{B}_R(\mathbb{R}^d)), \quad \langle \alpha, \mathcal{R}\varphi \rangle = \int_{\mathcal{B}_R(\mathbb{R}^d))} f(x) \Delta\varphi(x)\, dx, \tag{13}$$

$$\forall k, j, k' \in \mathbb{Z}^+ \text{ s.t. } k' \equiv k \ (mod\ 2),\ k' < k,\ 1 \le j \le N_{k,d},$$
$$\langle \alpha, Y_{k,j} \otimes X^{k'} \rangle = -c_d \langle f, (-\Delta)^{(d+1)/2} \mathcal{R}^*(Y_{k,j} \otimes \mathbb{1}_{|X| < R} X^{k'}) \rangle. \tag{14}$$

*The condition* (13) *holds for any measure $\alpha' \in \mathcal{M}(\mathbb{P}_R^d)$ for which $f$ admits a representation of the form* (7) *on $\mathcal{B}_R(\mathbb{R}^d)$. Thus, $\alpha$ can be characterized as the unique measure in $\mathcal{M}(\mathbb{P}_R^d)$ such that $f$ admits a representation of the form* (7) *on $\mathcal{B}_R(\mathbb{R}^d)$ and the condition* (14) *holds.*

In (14), the quantity $\langle f, (-\Delta)^{(d+1)/2} \mathcal{R}^*(Y_{k,j} \otimes \mathbb{1}_{|X| < R} X^{k'}) \rangle$ is well defined despite $\mathbb{1}_{|X| < R} X^{k'}$ not being continuous on $\mathbb{R}$; we define it as $\langle f, (-\Delta)^{(d+1)/2} \mathcal{R}^*((Y_{k,j} \otimes \mathbb{1}_{|X| < R} X^{k'}) + \tilde{g}) \rangle$, where $\tilde{g}$ is any function in $\mathcal{S}(\mathbb{P}_{R'}^d)$ such that $(Y_{k,j} \otimes \mathbb{1}_{|X| < R} X^{k'}) + \tilde{g} \in \mathcal{S}(\mathbb{P}_{R'}^d)$ (which do exist, see App. C).

In short, Proposition 4 characterizes the measure $\alpha$ from Proposition 1 in terms of its evaluations on the spaces $\mathcal{R}(\mathcal{S}(\mathcal{B}_R(\mathbb{R}^d)))$ and $\mathrm{span}(B)$, and by Corollary 1 the direct sum of these two spaces dense in $C_0(\mathbb{P}_R^d)$, which by the Riesz-Markov-Kakutani representation theorem is the predual of $\mathcal{M}(\mathbb{P}_R^d)$. Interestingly, the condition (13) holds for any measure $\alpha \in \mathcal{M}(\mathbb{P}_R^d)$ which represents the function $f$ on $\mathcal{B}_R(\mathbb{R}^d)$, but it is easy to see that the condition (14) does not: by Theorem 5 we have that if $\psi \in \mathrm{span}(A) \subseteq \mathrm{span}(B)$, the measure $\alpha'$ defined as $d\alpha'(\omega, b) = d\alpha(\omega, b) + \psi(\omega, b)\, db$ represents the function $f$ on $\mathcal{B}_R(\mathbb{R}^d)$, and $\langle \alpha', \psi \rangle = \langle \alpha, \psi \rangle + \|\psi\|_2^2$.

It remains an open question to see whether Theorem 5 captures all the measures which represent the zero function on $\mathcal{B}_R(\mathbb{R}^d)$, which we hypothesize. If that was the case, we would obtain a complete characterization of the Radon measures which represent a given function on $\mathcal{B}_R(\mathbb{R}^d)$.

**Mode connectivity**. Mode connectivity is the phenomenon that optima of neural network losses (at least the ones found by gradient descent) turn out to be connected by paths where the loss value is almost constant, and was observed empirically by Garipov et al. (2018); Draxler et al. (2018). Kuditipudi et al. (2019) provided an algorithmic explanation based on dropout, and an explanation based on the noise stability property. Theorem 5 suggests an explanation for mode connectivity from a functional perspective: one can construct finitely-supported measures which approximate a measure $\alpha \in \mathrm{cl}_w(\mathrm{span}(A))$, yielding finite-width neural networks with non-zero weights which approximate the zero function on $\mathcal{B}_R(\mathbb{R}^d)$. Assuming that the data distribution is supported in $\mathcal{B}_R(\mathbb{R}^d)$, adding a multiple of one such network to an optimal network will produce little change in the loss value because the function being represented is essentially unchanged. More work is required to confirm or discard this intuition.

## 6 CONCLUSION

We provided in this paper tighter sparse approximation bounds for two-layer ReLU neural networks. Our results build on the introduction of Radon-based $\mathcal{R}, \mathcal{U}$-norms for functions defined on a bounded open set $\mathcal{U}$. Our bounds refine Fourier-based approximation bounds of Breiman (1993); Klusowski & Barron (2018). We also showed that the representation of infinite width neural networks on bounded open sets are not unique, which can be seen as a functional view of mode connectivity observed in training deep neural networks. We leave two open questions: whether any function admitting an infinite-width representation on $\mathcal{U}$ has a finite $\mathcal{R}, \mathcal{U}$-norm, and whether Theorem 5 captures all the measures which represent the zero function on $\mathcal{B}_R(\mathbb{R}^d)$. Finally, in order to extend our theory to deeper ReLU networks we believe that non-linear Radon transforms (Ehrenpreis, 2003) are interesting tools to explore.

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
