# OpenReview forum: "Tighter Sparse Approximation Bounds for ReLU Neural Networks"
_ICLR.cc/2022/Conference — ICLR 2022 Spotlight_

### Official Review · Reviewer_2kZP · 2021-10-21

**Correctness:** 4
**Technical Novelty And Significance:** 3
**Empirical Novelty And Significance:** 3
**Recommendation:** 6
**Confidence:** 4

**Main Review:**

The novelty of this paper is high. By the introduction of R, U-norms the authors show that their approximation bound is tighter than bounds in the previous papers, and meaningful in more instances such as finite-width neural networks. This result will certainly help us have a better understating of deep neural networks from a theoretical way. I have checked the technique parts and find that the proofs are solid. I think this is a significant contribution to the deep learning community. The theoretical results in this paper are about two-layer ReLU neural networks. It will be interesting to see extended results on more widely used architectures in the future. Overall,  I think the results in this paper are important, as explained above.

**Summary Of The Paper:**

This paper studies the class of functions that coincide with an infinite width two-layer neural network on a fixed bounded open set. First, they introduce a Radon-based R, U-norms for functions defined on a bounded open set U. Then they prove tighter sparse approximation bounds for two-layer ReLU neural networks. They also prove that the representation of infinite width neural networks on bounded open sets are not unique.

**Summary Of The Review:**

By the introduction of R, U-norms the authors show that their approximation bound is tighter than bounds in the previous papers, and meaningful in more instances such as finite-width neural networks. The novelty of this paper is high. Overall,  I think the results in this paper are significant.

---

> ### Author Response · Authors · 2021-11-12
> **Response to Reviewer 2kZP**
>
> We want to thank the reviewer for their review and are pleased about their favorable view of our paper. We remain at their disposal to respond to any further remarks.

---

### Official Review · Reviewer_tNiY · 2021-10-30

**Correctness:** 4
**Technical Novelty And Significance:** 3
**Empirical Novelty And Significance:** Not applicable
**Recommendation:** 8
**Confidence:** 3

**Main Review:**

I recommend acceptance.

STRENGTHS: The Radon semi-norm is a potentially useful way of understanding representation, and as the paper shows, there are examples where it can provide a much better bound than the Barron norm.

WEAKNESSES: None that I am aware of.

QUESTIONS:
- Are there examples where ||_{R,U} is substantially smaller than ||_R (e.g. is this true for Example 1)? Should we expect it to be much smaller for many reasonable functions? More discussion on when/why this bound is better than the prior bounds would be helpful.

**Summary Of The Paper:**

The main result of this paper is a representation theorem for functions on a bounded set U by two-layer neural networks with bounded width and bounded weights, via a semi-norm ||_{R,U} based on the Radon transform. Prior work [OWSS19] used the Radon transform to get a representation theorem for functions on R^d, but the bounded-domain setting is more realistic and allows for stronger results, since the semi-norm ||_{R,R^d} will naturally be larger than ||_{R,U}. Moreover, this paper shows that when U is a Euclidean ball, the semi-norm ||_{R,U} can be upper bounded in terms of the Barron norm, so this paper's representation theorem strengthens Barron's Theorem.

**Summary Of The Review:**

This paper extends prior work on infinite-width NN approximability for bounded Radon semi-norm to finite-width NN approximability on bounded sets, which is a more reasonable setting and strengthens the classical Barron norm bounds.

---

> ### Author Response · Authors · 2021-11-12
> **Response to Reviewer tNiY**
>
> We thank the reviewer for their suggestions and are pleased about their favorable opinion of our work.
>
> **“Are there examples where $||_{R,U}$ is substantially smaller than $||_R$ (e.g. is this true for Example 1)? Should we expect it to be much smaller for many reasonable functions?”:**
>
> This is a very good question. Beyond the particular function in Example 1 that you point out, Theorem 4 allows us to answer it at a higher level of generality: when $f$ admits the modified Fourier representation (10), we have that
> $\|f\|_{\mathcal{R},\mathbb{B}_R(\mathbb{R}^d)}=$
>
> $\int_{-R}^{R} \int_{\mathbb{S}^{d-1}}  |\int_{\mathbb{R}} t^2 e^{-itb} \, d\mu(\omega,t)| db.$
>
>
> Since $b \to \int_{\mathbb{S}^{d-1}} |\int_{\mathbb{R}} t^2 e^{-itb} \, d\mu(\omega,t)|$ is a non-negative function, it is easy to see that $\|f\|_{\mathcal{R},\mathbb{B}_R(\mathbb{R}^d)}$ will be non-decreasing in $R$. That is, for larger balls the $\mathcal{R},\mathbb{B}_R(\mathbb{R}^d)$-norm of $f$ will be larger, and its limit when $R$ goes to infinity, which is equal to the $\mathcal{R}$-norm of $f$, may or may not be infinite.
>
> In fact, the computation of the $\mathcal{R},\mathcal{U}$-norm in Example 1 follows from Theorem 4 and we can see that in this case the quantity $\|f\|_{\mathcal{R},\mathbb{B}_R(\mathbb{R}^d)}$ increases with $R$, and that it tends to infinity as $R$ tends to infinity (which means that the $\mathcal{R}$-norm of $f$ is infinite).
>
> Namely, if we go to the proof of Example 1, we see that $\|f\|_{\mathcal{R},\mathbb{B}_R(\mathbb{R}^d)} =$
>
> $ \int_{-R}^{R} |-cos(x) + (1+\epsilon)^2 cos((1+\epsilon) x)| dx$. A quick plot of the function $x \to -cos(x) + (1+\epsilon)^2 cos((1+\epsilon) x)$ with small $\epsilon > 0$ shows that it is a sinusoidal with amplitude proportional to $|x|$, from which the statements follow.
>
> We include this discussion in Section 2.1.
>
> PS: Sorry about the formatting of some of the equations in the rebuttal, we encountered some issues in markdown rendering of equations.

---

### Official Review · Reviewer_3amN · 2021-11-02

**Correctness:** 4
**Technical Novelty And Significance:** 3
**Empirical Novelty And Significance:** Not applicable
**Recommendation:** 6
**Confidence:** 2

**Main Review:**

Strength:

This paper uses the novel analysis tool based on Ongie et al. (2019) to consider the bounded open set case. It is technically solid. I check the detailed proof and don't find the major flaws. (I need to admit I am not very familiar with the Fourier-based approximation bounds, so it is possible my judgment on novelty is wrong. And this is why I rate my confidence as 2.)

Weakness:

Clarify: It is still room to improve the technical part of the paper. 5 theorems and 4 propositions are shown in the 9-page main paper and more discussions on the Theorems/Proposition may be needed to make the paper easier to follow. I also suggest the authors add some discussions to further highlight the technical difference/significance of the Theorem/Proposition.


**Summary Of The Paper:**

This paper studies the approximation bounds for the 2-layer  ReLU network.  The authors extend the analysis framework of Ongie et al. (2019) and show the approximation bounds for the infinite-width network on a bounded open set as well as the bounds for sparse (i.e., finite-width) network that refine the similar results in the literature. At last, the authors show that the infinite-width neural network representations may not be unique on bounded open sets and provide a functional view of the mode connectivity. In particular, the authors refine the R-norm introduced by Ongie et al. (2019) to R, U-norm to tickle the bounded open set case.



**Summary Of The Review:**

This paper uses the novel analysis tool to study the approximation bounds for 2-layers ReLU networks and is technically solid. However, the presentation of the paper is a little bit hard to follow and I've spent about 15 hours checking technical details of the main paper and supplyments...

---

> ### Author Response · Authors · 2021-11-12
> **Response to Reviewer 3amN**
>
> We thank the reviewer for their questions and suggestions.
>
> **“...more discussions on the Theorems/Proposition may be needed to make the paper easier to follow.”**
>
> We tried to contextualize each of our results in order to underline their relevance and applicability, as well as their connection to previous work. We understand that the paper may not be easily accessible to readers unfamiliar with measure-theoretic tools. If the reviewer thinks that a particular result or section needs further clarification, we will be happy to work on it, within the space constraints.
>
> **“I also suggest the authors add some discussions to further highlight the technical difference/significance of the Theorem/Proposition.”**
>
> In the following we detail the criterion to decide whether we call a result a Theorem or a Proposition. As is customary, Propositions refer to results of tactical nature, that may be valuable on their own and for future works, but are primarily stated either as an intermediate step for subsequent developments, or as a digression from the key points of the paper. Theorems are the results where the main messages of the paper are delivered, and supposedly those that will be more useful to other researchers in the field.

---

### Official Review · Reviewer_E6s2 · 2021-11-05

**Correctness:** 4
**Technical Novelty And Significance:** 3
**Empirical Novelty And Significance:** Not applicable
**Recommendation:** 6
**Confidence:** 3

**Main Review:**

The presented tighter sparse approximation bounds for two-layer ReLU neural networks are of interest. Theorem 1 tells that there exists a neural network representation for a function defined on a bounded open set $\mathcal{U}$ if its $\mathcal{R}$, $\mathcal{U}$-norm is finite, which generalizes the results for functions on $\mathbb{R}^d$ in the literature and yields the sparse approximation bound in Theorem 2. Moreover, the authors discuss its tightness and links with Fourier sparse approximation bounds in Theorems 3 and 4. The non-uniqueness of neural network representations is studied in Theorem 5. I have one concern that how to check the finiteness of the $\mathcal{R}$, $\mathcal{U}$-norm for general functions?


Other minor comments/typos:
- rows above (1) and (2). Should $n$ be used instead of $m$?
-Proposition 2, $n$ or $m$ in the approximation bound?


**Summary Of The Paper:**

The paper provides some interesting and tighter spare approximation bounds for two-layer ReLU neural networks. The authors generalize the results for $\mathbb{R}^d$ to a bounded open set $\mathcal{U}\subset\mathbb{R}^d$ by defining the Radon-based $\mathcal{R},\mathcal{U}$-norms of functions. They also show that the representation of infinite width neural networks on $\mathcal{U}$ are not unique.

**Summary Of The Review:**

The paper is well organized and technically sound. The results should be interesting to the community of theoretical deep learning.

---

> ### Author Response · Authors · 2021-11-12
> **Response to Reviewer E6s2**
>
> We thank the reviewer for their suggestions.
>
> **“I have one concern that how to check the finiteness of the R,U-norm for general functions?”**
>
> The issue raised by the reviewer is clearly relevant with regards to the scope and applicability of our approximation bounds. While general functions may not have a finite $\mathcal{R},\mathcal{U}$-norm, in the paper we provide two ways two check that a function has a finite $\mathcal{R},\mathcal{U}$-norm:
> In Theorem 3, we show that if the quantity $C_f$ (defined in Sec. 2.2 in terms of the Fourier transform of $f$) is finite, then the $\mathcal{R},\mathcal{U}$-norm is finite and can be upper-bounded in terms of $C_f$. Furthermore, in Theorem 4 we give an expression of the $\mathcal{R},\mathcal{U}$-norm in terms of the modified Fourier representation from equation 10.
> The second class of functions for which we prove finiteness of the $\mathcal{R},\mathcal{U}$-norm is the class of two-layer neural networks with finite width (Proposition 3). Interestingly, for this class of functions the quantities $C_f$ and $\tilde{C}_f$ are generally infinite, which means that our Radon-based sparsity bounds apply in a strictly wider range of settings than the classical Fourier-based sparsity bounds introduced in Sec. 2.2.
>
> **Response to minor comments and typos:**
>
> In rows above (1) and (2), indeed $n$ should be used instead of $m$.
> In Proposition 2, it should also be $n$ instead of $m$.
> We want to thank the reviewer for catching these two typos, we will correct them.

---

### Decision · Program_Chairs · 2022-01-20

**Decision:**

Accept (Spotlight)

**Comment:**

The authors extend the result of Ongie et al. (2019)  and derive sparseneural network approximation bounds that refine previous results. The reuslts are quite ineteresting and relevant to ICLR. All the reviewers were positive about this paper.